# A Systematic Review and Meta-Analysis of the Prognostic Impact of Pretreatment Fluorodeoxyglucose Positron Emission Tomography/Computed Tomography Parameters in Patients with Locally Advanced Cervical Cancer Treated with Concomitant Chemoradiotherapy

**DOI:** 10.3390/diagnostics11071258

**Published:** 2021-07-14

**Authors:** Lu Han, Qi Wang, Lanbo Zhao, Xue Feng, Yiran Wang, Yuliang Zou, Qiling Li

**Affiliations:** 1Department of Obstetrics and Gynecology, First Affiliated Hospital, Xi’an Jiaotong University, Xi’an 710061, China; hanlu@stu.xjtu.edu.cn (L.H.); wangqiw@stu.xjtu.edu.cn (Q.W.); boboupzu2@stu.xjtu.edu.cn (L.Z.); fengxue1995214@stu.xjtu.edu.cn (X.F.); 15529519211@163.com (Y.W.); zouyuliangfl@126.com (Y.Z.); 2Department of Gynecological Oncology, Shaanxi Provincial Tumor Hospital, Xi’an 710065, China

**Keywords:** positron-emission tomography, prognosis, uterine cervical neoplasms, meta-analysis, chemoradiotherapy

## Abstract

Backgrounds: The purpose of this paper is to investigate the prognostic value of fluorodeoxyglucose positron emission tomography/computed tomography (FDG PET/CT) parameters in patients treated with concurrent chemoradiotherapy (CCRT) for locally advanced cervical cancer (LACC). Methods: Studies that met the following criteria were retrieved from PubMed and Embase: patients treated with CCRT for LACC; FDG PET/CT scans performed before CCRT treatment; and a detected relationship between the parameters of FDG PET/CT and the prognosis of patients. Pooled hazard ratios (HRs) with 95% confidence intervals (CIs) were used to estimate the overall survival (OS) or event-free survival (EFS). Results: In total, 14 eligible studies with 1313 patients were included in this meta-analysis. Patients with a high maximum standardized uptake value (SUVmax) have a shorter OS than those with a low SUV_max_ (HR = 2.582, 95% = CI 1.936–3.443, *p* < 0.001). Primary tumor SUV_max_ values (HR = 1.938, 95% CI = 1.203–3.054, *p* = 0.004) were significantly correlated with EFS, with a relatively high heterogeneity (*I*^2^ = 84% and *I*^2^ = 69.4%, respectively). Based on the limited data, the combined HR for EFS with the highest primary tumor total lesion glycolysis (TLG) and metabolic tumor volume (MTV) was 1.843 (95% CI = 1.100–3.086, *p* = 0.02) and 2.06 (95% CI = 1.21–3.51, *p* = 0.007), respectively. Besides, the combined HR for OS with the highest nodal SUV_max_ was 2.095 (95% CI = 2.027–2.166, *p* < 0.001). Conclusion: A high primary SUVmax has a significant correlation with the OS and EFS of patients treated with CCRT for LACC and may therefore serve as a prognostic predictor. Due to the limited data, to explore the correlation between survival and TLG, MTV, and nodal SUVmax, further large-scale prospective studies are needed.

## 1. Introduction

Cervical cancer is a common gynecological malignancy, with around 569,847 new cases and 311,365 deaths annually worldwide, and a significant proportion of patients are diagnosed at a locally advanced stage [1]. For locally advanced cervical cancer (LACC), concurrent chemoradiotherapy (CCRT), using cisplatin-based chemotherapy in association with pelvic external-beam radiotherapy and subsequent brachytherapy, is considered the standard treatment [2]. While the contribution of CCRT toward an improvement in survival outcomes and reduction in recurrence has been confirmed, the complete clinical response of these patients is 70–90%, and about one-third of patients experience recurrence [3,4]. 

The accurate prediction of recurrence is of great significance. Several traditional clinical and pathological factors are identified as poor prognostic factors, which include the advanced International Federation of Gynecology and Obstetrics (FIGO) stage, presence of lymph node metastasis, parametrial invasion, histological tumor type, and a large tumor diameter [5,6,7]. However, there are still no satisfactory parameters sufficient to predict prognosis accurately.

Computed tomography (CT), magnetic resonance imaging (MRI), and fluorodeoxyglucose positron emission tomography/computed tomography (FDG PET/CT) are the most commonly used imaging modalities for CC. Compared to CT or MRI, FDG PET/CT could show metabolic information on tumors and more accurately assess lymph node involvement, distant metastasis, and recurrent disease [8]. Therefore, FDG PET/CT has been widely used in staging, therapeutic strategies, and the treatment response assessment of patients with CC [9]. 

With the technology developed, imaging has provided the potential for prognostic biomarkers [10]. Except for the aforementioned roles, quantitative parameters derived from FDG-PET/CT, including the maximum standardized uptake value (SUVmax), average SUV (SUVmean), metabolic tumor volume (MTV), and total lesion glycolysis (TLG), have recently been proposed as prognostic biomarkers for patients with LACC who are treated with CCRT. However, the results of some studies show some differences. For example, Im et al. detected that patients with a high SUV_max_ measurement in tumor tissue had a higher risk of recurrence or progression than those with a low SUV_max_ (hazard ratio [HR] = 2.14, 95%CI = 1.08–4.22, *p* = 0.029) [11], but Chong et al. drew the opposite conclusion (HR = 0.673, 95%CI = 0.5–4.0, *p* = 0.412) [12]. Therefore, we performed this meta-analysis to synthetize the relevance of FDG PET/CT parameters as prognostic biomarkers for patients with LACC who are treated with CCRT.

## 2. Methods

### 2.1. Literature Search and Selection Criteria

Embase and PubMed (from inception to December 2020) were systematically searched using the appropriate terminology, as described in Appendix A. Besides, the reference lists of the articles reviewed as full texts were also searched manually. The inclusion criteria in the meta-analysis were as follows: patients treated with CCRT for LACC; FDG PET/CT scans performed before or during CCRT treatment; and a detected relationship between the parameters of FDG PET/CT and the prognosis of patients. Studies meeting the following criteria were excluded: publication type other than original research articles (e.g., review articles, conference abstracts, or editorial), patients treated with surgery at any point in the treatment, patients treated with chemotherapy or radiotherapy alone, and patients lacking important information for analysis (e.g., articles reporting HR, using Cox proportional hazards modeling with parameters as a continuous variable). Two authors conducted the searches and screening independently. Any discrepancies were resolved by consensus with another reviewer.

### 2.2. Data Extraction 

The following information was extracted using a standardized form: surname of first author, year of publication, country, design (prospective or retrospective), patients’ characteristics (including patients’ number, tumor node metastasis [TNM] staging, histology, radiotherapy technique, the schedule of systemic therapy, and follow-up period), and outcomes.

### 2.3. Statistical Analysis 

The primary outcome, progression event-free survival (EFS), was defined as the date from initiating therapy to recurrence or metastasis. In some of the included studies, disease-free survival (DFS) or progression-free survival (PFS) was obtained as the primary outcome, but they were all redefined as EFS in this meta-analysis. The secondary outcome was overall survival (OS), defined as the date from initiating therapy until death. The effect sizes of the prognostic values of SUV_max_, SUV_mean_, MTV, and TLG were measured in terms of hazard ratio (HR). An HR greater than 1 implied a worse survival for patients with a high SUV_max_, SUV_mean_, MTV, or TLG. The most adjusted estimate of HR was extracted directly from each study, if provided by the authors. Otherwise, the HR estimate and its variance were extracted from Kaplan-Meier curves by the Engauge Digitizer, version 3.0 (http://digitizer.sourceforge.net, accessed date: 30 December 2020). The heterogeneity was assessed with the Q test and *I*^2^ statistic, and a *p* value less than 0.05 or *I*^2^ values higher than 50% indicated a significant heterogeneity. The fixed-effects model was used to estimate the cases with homogeneity, and the random-effects model was used for the cases with a significant heterogeneity. The publication bias of the studies was visually displayed by the asymmetry of an inverted funnel plot and quantitatively evaluated by Egger’s tests, with a *p* value less than 0.05 suggesting a significant publication bias. Sensitivity analyses were conducted to assess the stability of the meta-analysis results. All the statistical analyses were conducted using the STATA 14.0 (STATA Corporation, College Station, TX, USA) software.

## 3. Results 

### 3.1. Study Selection

We picked up 822 potentially eligible articles in the electronic search, with 88 articles removed for duplication and 644 articles excluded after the reviewers read the titles and abstracts. In total, 14 eligible studies with a total of 1313 patients were included in this meta-analysis, after reviewing the full text (Figure 1).

### 3.2. Patient Characteristics and Treatment Features

The median age of the patients included in the studies was 51.5–58.0 years (range from 21 to 89 years). Only 12 of the 14 studies provided information about histology, and 89.14% patients had a squamous cell carcinoma (1076/1205) (Table 1). The largest stage subgroup was the FIGO stage ⅡB, which was the case for 64.66% (846/1313) of the patients. The median follow-up for all the patients was 22–60 months (range from 3 to 129 months) (Table 2). No information on the radiotherapy technique used was available for the three studies, and most of the patients from the other studies were treated with 3D-conformal radiotherapy. The widely adopted external beam radiotherapy regimen consisted of conventional fractionation of 1.8 Gy per fraction (8/9 studies) for a total dose of 45–50.4 Gy. The intracavitary brachytherapy regimen used in the majority of patients consisted of a fractionation of 7 Gy per fraction (6/10 studies), with four cycles (4/9 studies). The adopted concurrent chemotherapy regimen was a standard weekly 40 mg/m^2^ cisplatin in most patients (10/13 studies) (Table 3).

### 3.3. Primary Outcome: EFS

Nine articles on SUV_max_ in tumor tissue were included in the study [11,12,14,15,16,17,18,21,23]. The combined HR for EFS with the highest SUV_max_ was 1.938 (95% CI = 1.203–3.054, *p* = 0.004) (Appendix A), which meant that the patients with a high SUV_max_ measurement in tumor tissue had a higher risk of recurrence or progression than those with a low SUV_max_. However, a significant heterogeneity existed between the articles (*I*^2^ = 84%, *p* = 0.000). The publication bias was significant based on the funnel graph (Appendix A) and Egger’s test (*p* = 0.015). The results obtained through the sensitivity analysis was relatively stable (Appendix A).

Four articles on primary tumor SUV_mean_ were included in the study [16,17,18,19]. Due to there being no significant heterogeneity (*I*^2^ = 0%, *p* = 0.938), the fixed-effects model was performed. The combined HR for EFS with the primary tumor SUV_mean_ was 1.182 (95% CI = 0.899–1.544, *p* = 0.230) (Appendix A). The Begg’s funnel plot and Egger’s test indicated no publication bias (*p* = 0.345), and the sensitivity analysis showed that the analysis was relatively stable (Appendix A).

Three articles on MTV (measuring tumor tissue) were included in the study [12,16,18]. With no significant heterogeneity (*I*^2^ = 20.9%, *p* = 0.282), the combined HR for EFS with the highest MTV was 2.06 (95% CI = 1.21–3.51, *p* = 0.007) (Figure 2A). This indicated that patients with a high MTV had a higher risk of recurrence or progression than those with a low MTV. Because the data were limited, the publication bias was not evaluated.

Four articles on primary tumor TLG were included in the study [13,16,18,19]. In the absence of a significant heterogeneity (*I*^2^ = 0%, *p* = 0.552), the fixed-effects model was applied. The combined HR for EFS with the primary tumor TLG was 1.843 (95% CI = 1.100–3.086, *p* = 0.02). (Figure 2B). The Begg’s funnel plot and Egger’s test indicated no publication bias (*p* = 0.261), and the sensitivity analysis showed that the analysis was relatively stable (Figure 2C,D).

Five articles on nodal SUV_max_ were included in the study [13,14,20,22,23]. Where there was a significant heterogeneity (*I*^2^ = 69.4%, *p* = 0.011), the random-effects model was applied. The combined HR for EFS with the nodal SUV_max_ was 3.478(95% CI = 2.006–6.029, *p* < 0.001) (Appendix A). The publication bias was significant based on the funnel graph (Appendix A) and Egger’s test (*p* = 0.04). The sensitivity analysis proved that the results were relatively stable (Appendix A).

### 3.4. Second Outcome: OS

Seven articles on SUV_max_ in tumor tissue were included in the study [11,14,15,21,23,24,25]. Where there was a moderate heterogeneity (*I*^2^ = 36.2%, *p* = 0.151), the fixed-effects model was applied when pooling the HR. The combined HR was 2.582 (95% CI = 1.936–3.443, *p* < 0.001), which meant that patients with a high SUV_max_ had a shorter survival time than those with a low SUV_max_ (Figure 3A). Begg’s funnel plot and Egger’s test indicated no publication bias (*p* = 0.113), and the sensitivity analysis showed that the analysis was relatively stable (Figure 3B,C).

Two articles on primary tumor SUV_mean_ were included in the study [15,19]. With no significant heterogeneity (*I*^2^ = 0.0%, *p* = 0.893), the combined HR for EFS with the highest MTV was 1.523 (95% CI = 0.764–3.038, *p* = 0.232) (Appendix A). Because of the limited data, the publication bias and sensitivity analysis were not performed.

Two articles on TLG in tumor tissue were included in the study [15,19]. In the absence of heterogeneity (*I*^2^ = 0.0%, *p* = 0.672), the fixed-effects model was applied when pooling the HR. The combined HR for EFS with the nodal SUV_max_ was 1.371 (95% CI = 0.681–2.759, *p* = 0.377) (Appendix A). Because of the limited data, the publication bias and sensitivity analysis were not performed.

Two articles on nodal SUV_max_ were included in the study [20,23]. Where there was no heterogeneity (*I*^2^ = 54.1%, *p* = 0.140), the fixed-effects model was applied when pooling the HR. The combined HR for EFS with the nodal SUV_max_ was 2.095 (95% CI = 2.027–2.166, *p* < 0.001) (Appendix A). Because of the limited data, the publication bias and sensitivity analysis were not performed.

## 4. Discussion

In this system review and meta-analysis, we review the prognostic value of primary tumor SUV_max,_ SUV_mean_, MTV, TLG, and nodal SUV_max_ in patients treated with CCRT for LACC. Besides, the results of meta-analysis show that patients with a high primary SUVmax have a shorter OS and a higher risk of recurrence or progression than those with a low SUV_max_. The synthesized results showed that the highest TLG and MTV were correlated with EFS, and the highest Nodal SUV_max_ was associated with OS in patients treated with CCRT for LACC. Due to the limited data, a publication bias, and heterogeneity, to explore whether MTV, TLG, and nodal SUV_max_ are significant markers and can act as prognostic indicators in clinical practice, further large-scale prospective studies are required.

This meta-analysis focused on the FDG PET/CT parameters as a categorical variable, but the results of the articles concerning FDG PET/CT parameters as a continuous variable exhibited an extreme discrepancy among different studies. For example, Carpenter et al. reported that primary tumor MTV and primary tumor TLG were significantly associated with OS in univariate analyses, whereas only primary tumor TLG were significantly correlated with OS in multivariate analyses [26]. Calles-Sastre et al. found that primary-tumor MTV and primary-tumor TLG showed a significant association with OS and with RFS in univariate analyses [27]. Scher et al. found that primary-tumor MTV and primary-tumor TLG were correlated with OS and DFS in univariate analyses, whereas primary-tumor TLG was the only FDG PET parameter significantly correlated with OS and DFS in multivariate analyses [25]. In addition, Liu et al. confirmed that primary-tumor MTV was a significant predictor of OS in univariate analyses [28]. Chong et al. reported that primary-tumor SUV_max_, primary tumor MTV, and primary tumor TLG were significant prognostic factors for DFS in univariate analyses, but none of them was significant in multivariate analyses (Appendix A) [22]. 

Apart from pretreatment FDG PET/CT parameters, some studies focused on the prognostic value of during-treatment FDG PET/CT parameters. Carpenter et al. reported that primary-tumor MTV and primary-tumor TLG during treatment were significantly associated with OS and DFS [26]. Liu et al. found that primary-tumor MTV during treatment was significantly associated with OS [28]. Krhili et al. confirmed that not only primary tumor MTV and primary-tumor TLG, but also SUV_max_ during treatment were significant prognostic factors for OS and DFS (Appendix A) [29]. Besides, changes between pre- and intra-treatment FDG-PET/CT parameters were also confirmed to be prognostic factors in some studies. Park et al. confirmed that the percentage changes of SUV_max_ have a prognostic value for predicting DFS [30]. Oh et al. showed that a decrease of SUV_max_ was a statistically significant predictor of PFS [31]. However, Carpenter et al. reported no correlation among the changes in primary-tumor SUV_max_, SUV_mean_, MTV, TLG, and survival [26]. 

Except for the pretreatment FDG PET/CT parameter, during-treatment FDG PET/CT parameter, and changes between pre- and intra-treatment FDG-PET/CT parameters, the one-two punch, using various types of prognostic factors to create a prognosis-predicting model, was a new orientation. For example, Hong et al. suggested a simple prognosis prediction model, using pretreatment FDG PET/CT and human papillomavirus (HPV) genotyping in patients with LACC treated with CCRT [32]. Lee et al. constructed a nomogram based on these six (including age, nodal SUV_max_, primary-tumor SUV_max_, size, stage, and SCC) and seven (including age, nodal SUV_max_, primary-tumor SUV_max_, size, stage, SCC, and high-risk HPV status) independent risk factors for two-year DFS and five-year OS, respectively [23]. Besides, radiomics is a new frontier for predicting the prognosis of patients with LACC. For example, Lucia et al. found that in LACC treated with CCRT, radiomics features, such as Grey Level Non-uniformity GLRLM from PET, are independent predictors of recurrence and loco-regional control with a significantly higher prognostic power than usual clinical parameters [17]. The predictive tool-enrolled FDG PET/CT parameter still needs more study. 

Our study is the first system review and meta-analysis to evaluate the prognostic value of FDG PET/CT metabolic parameters (including primary-tumor SUV_max_, SUV_mean_, MTV, TLG, and nodal SUV_max_) in patients with LACC treated with CCRT. We performed an extensive search addressing all the available databases, and our meta-analysis confirmed the significant association with the primary-tumor SUV_max_ and OS of patients with LACC treated with CCRT. Future studies could focus on the prognosis-predicting model based on the primary-tumor SUV_max_ using various types of prognostic factors. However, there were still some noteworthy limitations of this meta-analysis. First, the small article scale concerning TLG, MTV, and SUV_max_ did not support a high-quality and valuable conclusion, and large-scale prospective studies are urgently need to explore the clinical value of our findings. Besides, a significant heterogeneity existed in the results of this meta-analysis, and no subgroup analysis was performed for the limited studies. The potential source of the heterogeneity was probably four-fold: (1) The patients in each study had different FIGO stages, different histologic subtypes, different treatment strategies, and different follow-up endpoints (Table 1, Table 2 and Table 3); (2) different PET/CT techniques and image analyses existed in each study, which might have caused the heterogeneity in this analysis (Appendix A); (3) the cut-off values for FDG PET/CT metabolic parameters were different in each study; and (4) the HR used in this meta-analysis was directly extracted from a Cox proportional hazards model (multivariate analysis and univariate analysis) or indirectly estimated from Kaplan-Meier curves.

## 5. Conclusions

A high primary SUV_max_ has a significant association with a shorter OS and EFS in patients treated with CCRT for LACC, which might serve as a prognostic predictor. Due to the limited data, a publication bias, and heterogeneity, to explore the correlation between survival and TLG, MTV, and nodal SUV_max_, further large-scale studies are needed.

## Figures and Tables

**Figure 1 diagnostics-11-01258-f001:**
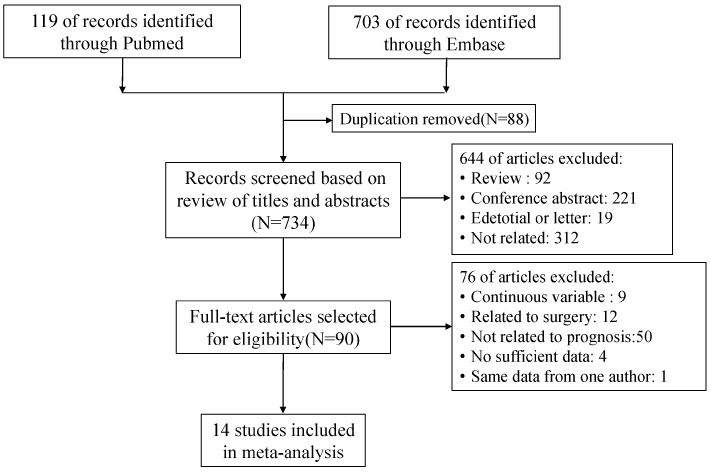
Flow chart of the literature search.

**Figure 2 diagnostics-11-01258-f002:**
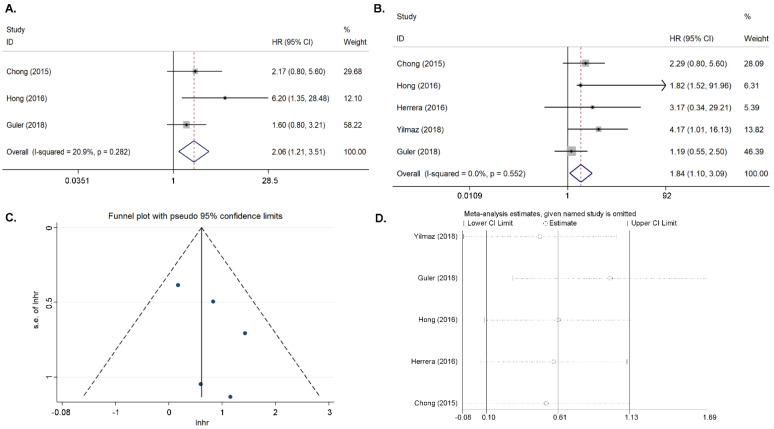
Results for the primary MTV and TLG for DFS. (**A**) Forest plots of the HRs for DFS with MTV. (**B**) Forest plots of the HRs for DFS with TLG. (**C**) Begg’s funnel plot for TLG. (**D**) Sensitivity analysis of the influence of each individual study on the pooled HR for TLG.

**Figure 3 diagnostics-11-01258-f003:**
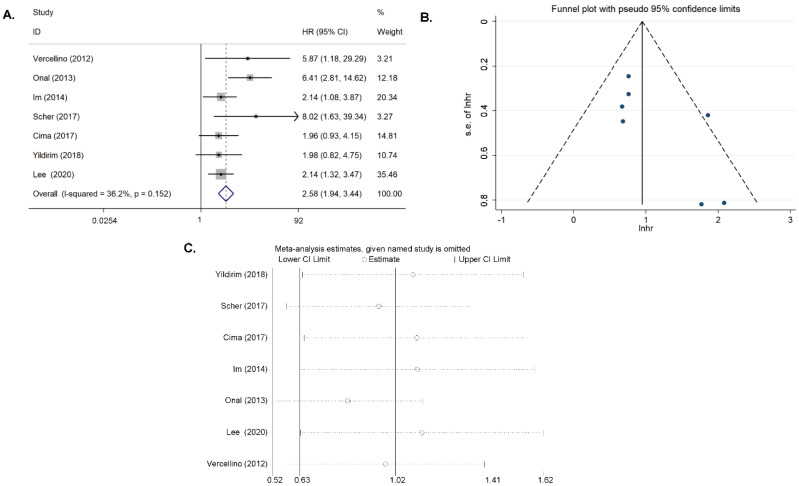
Results for the primary SUV_max_ for OS. (**A**) Forest plots of the HRs. (**B**) Begg’s funnel plot. (**C**) Sensitivity analysis of the influence of each individual study on the pooled HR.

**Table 1 diagnostics-11-01258-t001:** Basic information and patient characteristics.

Surname of First Author	Year	Country	Study Design	No. of Patients	Age		Histology		
					Median	Range	SSC	ACa	Other
Yilmaz [13]	2018	Turkey	Retrospective	44	54.6	28–78	NS	NS	NS
Yildirim [14]	2018	Turkey	Retrospective	63	58	21–86	NS	NS	NS
Cima [15]	2017	Switzerland	Retrospective	92	57	30–89	71	14	7
Guler [16]	2018	Turkey	Retrospective	129	57	22–83	119	10	-
Lucia [17]	2018	France	Retrospective	69	58	29–90	54	9	6
Hong [18]	2016	Korea	Retrospective	56	57	32–81	49	7	-
Herrera [19]	2016	Switzerland	Retrospective	38	52.5	26–83	33	5	-
Chong [12]	2015	Korea	Retrospective	56	51.5	NS	50	6	-
Onal [20]	2015	Korea	Retrospective	93	58	30–89	87	6	-
Im [11]	2014	Korea	Retrospective	145	55	22–88	131	10	4
Onal [21]	2013	Turkey	Retrospective	149	58	21–86	138	11	-
Chong [22]	2017	Korea	Retrospective	93	53.1	NS	85	5	-
Lee [23]	2020	Korea	Retrospective	270	NS	NS	243	NS	NS
Vercellino [24]	2012	France	Retrospective	16	57	32–69	15	1	NS

SSC, squamous cell carcinoma; AC, adenocarcinoma, NS, not stated.

**Table 2 diagnostics-11-01258-t002:** Disease features: stage and follow-up.

Surname of First Author	Stage						Follow-Up	
	ⅠB	ⅡA	ⅡB	ⅢA	ⅢB	Ⅳ	Median	Range
Yilmaz [13]	7	2	27	1	3	4	22	8–54
Yildirim [14]	4	4	38	8	9	-	60	3–125
Cima [15]	-	-	78	3	4	7	31	6–85
Guler [16]	16	4	52	13	36	8	30	3.7–94.7
Lucia [17]	10	4	36	1	9	9	36	6–79
Hong [18]	-	-	38	3	6	9	NS	NS
Herrera [19]	6	10	15	2	4	1	52.5	26–83
Chong [12]	-	-	44	7	5	-	42	6–97
Onal [20]	9	2	49	10	21	2	29	3–79
Im [11]	19	8	90	5	20	3	NS	NS
Onal [21]	16	7	84	15	24	3	29	3–79
Chong [22]	-	-	75	12	5	1	55	9–124
Lee [23]	-	-	215	NS	NS	NS	49.5	3–129
Vercellino [24]	1	1	8	3		3		

NS, not stated.

**Table 3 diagnostics-11-01258-t003:** Treatment-related features.

Surname of First Author	EBRT			ICR		Concurrent Chemotherapy
	Type	Total Dose Median (Range)	Fraction (Gy)	Fraction (Gy)	Number	Type
Yilmaz [13]	3D-CRT	50.4	1.8	6	5	Weekly 40 mg/m^2^ cisplatin
Yildirim [14]	3D-CRT	50.4	1.8	7	4	Weekly 40 mg/m^2^ cisplatin
Cima [15]	3D-CRT	NS	NS	NS	NS	Weekly 40 mg/m^2^ cisplatin
Guler [16]	3D-CRT	50.4	1.8	7	4	Weekly 40 mg/m^2^ cisplatin
Lucia [17]	3D-CRT/IMRT	45–50.4	NS	7	3–4	Weekly 40 mg/m^2^ cisplatin
Hong [18]	NS	54	2	3.5	8	Weekly 30 mg/m^2^ cisplatin or triweekly paclitaxel 135 mg/m^2^ and carboplatin
Herrera [19]	IGRT	45 (45–50.4)	NS	7	3–4	Weekly 40 mg/m^2^ cisplatin (89.4%)
Chong [12]	3D-CRT	45	1.8	-	-	Weekly 40 mg/m^2^ cisplatin
Onal [20]	3D-CRT	50.4	1.8	7	4	Weekly 40 mg/m^2^ cisplatin
Im [11]	NS	NS	NS	NS	NS	Weekly cisplatin
Onal [21]	3D-CRT	50.4	1.8	7	4	Weekly 40 mg/m^2^ cisplatin
Chong [22]	3D-CRT	45	1.8	6	NS	Weekly 40 mg/m^2^ cisplatin
Lee [23]	3D-CRT	45	1.8	NS	NS	Weekly 40 mg/m^2^ cisplatin
Vercellino [24]	NS	NS	NS	NS	NS	NS

EBRT, External Beam Radiotherapy; ICR Intracavitary Brachytherapy IGRT, Image-Guided Radiation Therapy; 3D-CRT, 3D Conformal Therapy; IMRT, Intensity-Modulated Radiation Therapy; NS, not stated.

## Data Availability

All the data used to support the findings of this study are included in the article.

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
