# Peer review of "A Systematic Review and Meta-Analysis of the Prognostic Impact of Pretreatment Fluorodeoxyglucose Positron Emission Tomography/Computed Tomography Parameters in Patients with Locally Advanced Cervical Cancer Treated with Concomitant Chemoradiotherapy"

_diagnostics, 2021, doi:10.3390/diagnostics11071258_

Round 1

Reviewer 1 Report

This is a well-written review about the value of pretreatment FDG-PET/CT in patients with locally advanced cervical cancer. In my opinion, in can be accepted for publication after minor english editing.

Author Response

Response to Reviewer 1 Comments

Point 1: This is a well-written review about the value of pretreatment FDG-PET/CT in patients with locally advanced cervical cancer. In my opinion, it can be accepted for publication after minor English editing.

Response 1: We appreciate your agreement and feel so sorry for the awkward English expressions, but we have tried our best to revise this paper many times and asked the grammar modification company for further polish. The certification of the grammar modification is attached at the end of this letter.

Reviewer 2 Report

This meta-analysis reviewed prognostic value of pretreatment FDG PET/CT in locally advanced cervical cancer. MTV and TLG of primary tumor were significant biomarker for predict to tumor recurrence.

Major comment

1. The authors concluded primary tumor SUVmax and nodal SUVmax were significant correlated with relatively high heterogeneity. Did you mean metabolic parameter with high heterogeneity have significant meaningful parameters clinically or not?

2. According to lymph node metastasis, the significance of primary tumor metabolic parameters may be altered. Did you have any meta-analysis of prognostic value of primary tumor metabolic parameters according to lymph node status?

Author Response

Response to Reviewer 2 Comments

Point 1: The authors concluded primary tumor SUVmax and nodal SUVmax were significant correlated with relatively high heterogeneity. Did you mean metabolic parameter with high heterogeneity have significant meaningful parameters clinically or not?

Response 1: We appreciate your valuable comment. The conclusion concerning associated between EFS and primary tumor SUVmax and nodal SUVmax existed significant heterogeneity in this article.

Five articles concerning the association between nodal SUVmax and EFS were included in this meta-analysis. All articles drew the same conclusion that patient with higher nodal SUVmax was a risk factor. Thus, we believed that the conclusion with some heterogeneity didn’t influence the clinical use. With more researches on the metabolic parameter, the heterogeneity could be eliminated by the more detailed subgroup analysis.

Nine articles concerning the association between primary tumor SUVmax and EFS were included in this meta-analysis. All articles drew the similar conclusion that higher SUVmax measurement on tumor tissue was a risk factor except for two studies with the opposite opinion, which cause the heterogeneity. First, SUVmax measurements are highly dependent on the scanner model, reconstruction algorithm and settings, post-filtering strategies, as well as other factors. Besides, patients in different studies with unequal age, stage, histology, therapy, and follow-up period. All the above-mentioned factors might influence the conclusion and cause the heterogeneity; therefore, we thoroughly collected the above factors from the enrolled study in the Tables of this article. With more researches on the metabolic parameter, the heterogeneity could be eliminated by the more detailed subgroup analysis.

Point 2: According to lymph node metastasis, the significance of primary tumor metabolic parameters may be altered. Did you have any meta-analysis of prognostic value of primary tumor metabolic parameters according to lymph node status?

Response 2: We extremely appreciate you for your valuable comment and for providing a significant research direction. Because no article provided the results about the prognostic value of primary tumor metabolic parameters according to lymph node status,and we do not have the original data about lymph node metastasis status and survival time of each patient in the article enrolled in this meta-analysis, we could provide the conclusion about the prognostic value of primary tumor metabolic parameters according to different lymph node status. We will ask each author for the original data and collect data about this part in our hospital in our next research.

Reviewer 3 Report

In this manuscript, the authors conducted a meta-analysis study concerning the prognostic performance of the metabolic parameters on FDG PET in advanced cervical cancer patients treated by chemoradiotherapy. Fourteen studies were included in the final analysis. Concerning event-free survival (EFS), nine articles about SUVmax were included. Three and four articles were used for metabolic tumor volume (MTV) and total lesion glycolysis (TLG), respectively. As for overall survival (OS), seven articles were included to investigate SUVmax and two articles about total lesion glycolysis were used.

In the result, the authors stated that higher primary tumor total lesion glycolysis (HR = 1.843, 95% CI = 1.100–3.086, P = 0.02) and metabolic tumor volume (HR = 2.06, 95% CI, 1.21–3.51, P = 0.007) were significantly associated with shorter EFS. Patients with high maximum standardized uptake value (SUVmax) have a shorter OS than those with low SUVmax (HR = 2.582, 95% CI 1.936–3.443, P <0.001). Primary tumor SUVmax (HR = 1.938, 95% CI, 1.203–3.054, P = 0.004) and nodal SUVmax (HR = 3.478, 95% CI = 2.006-6.029, P <0.001) were significantly correlated with EFS with relatively high heterogeneity (I2 = 84% and I2 = 69.4%, respectively).

They concluded that pretreatment FDG PET/CT parameters serve as a prognostic predictor.

(1) General consideration:

Originality/Novelty: Meta-analysis about the prognostic use of metabolic PET parameters in advanced cervical cancer patients had not been reported.

Significance of Content: All the articles are retrospective in nature. The HRs of SUVmax for OS and EFS range from 1.938 to 2.582. Not very high. The prognostic significance of PET parameters is moderate.

Quality of Presentation and Scientific Soundness: The main problem of the analysis is the number of articles about MTV and TLG is few. The publication bias cannot be well assessed using funnel plot in such low number. It is doubtful that why TLG and MTV were still reported to be significant factors after the meta-analysis. Only the number of articles about SUVmax is somewhat enough to draw a conclusion.

(2) Other points need explanation.

  1. In abstract, “Studies that met the following criteria were retrieved from PubMed and Embase: FDG PET/CT scans performed treatment” What is performed treatment?
  2. Table 1. Adding reference number of each article. I am not able to see where these articles were published?
  3. The first paragraph of Discussion. TLG and MTV are potential biomarkers for EFS? The low HR, few article numbers, and heterogeneity do not support your conclusion.
  4. Limitation of this manuscript. Few articles addressing TLG or MTV is a major limitation.

Author Response

Response to Reviewer 3 Comments

Point 1: In abstract, “Studies that met the following criteria were retrieved from PubMed and Embase: FDG PET/CT scans performed treatment” What is performed treatment?

Response 1: We are very sorry for this mistake and we change this part as " Studies that met the following criteria were retrieved from PubMed and Embase: patients treated with CCRT for LACC; FDG PET/CT scans performed before CCRT treatment; the relationship between parameters of FDG PET/CT and the prognosis of patients were investigated.”

Point 2: Table 1. Adding reference number of each article. I am not able to see where these articles were published?

Response 2: We are very sorry for the indistinct exhibition of the Table and we have rechecked and revised all Tables with the reference number of each article.

Point 3: The first paragraph of Discussion. TLG and MTV are potential biomarkers for EFS? The low HR, few article numbers, and heterogeneity do not support your conclusion.

Response 3: We are very sorry for the un-precise statement, and we have revised this wrong description as: “In this meta-analysis, the results revealed that MTV and TLG measurement of tumor tissue might be a biomarker for EFS, which need further be verified with large-scale prospective studies”. We are so sorry for the few article numbers and we searched the database again to expand the article scale. But it’s a pity that their no new relevant data about the prognostic value of TLG and MTV in patients with locally advanced cervical cancer treated with concomitant chemoradiotherapy. We hope that the research about the prognostic value of TLG and MTV will be encouraged by our review. We will keep an eye on this topic and collect some original data. Someday this is enough data we will perform a high-quality meta-analysis on the prognostic value of TLG and MTV in patients with locally advanced cervical cancer treated with concomitant chemoradiotherapy.

Point 4: Limitation of this manuscript. Few articles addressing TLG or MTV is a major limitation.

Response 4: We so sorry for this huge limitation. We searched the database again to expand the article scale. But it’s a pity that their no new relevant data about the prognostic value of TLG and MTV in patients with locally advanced cervical cancer treated with concomitant chemoradiotherapy. We have added this part in the discussion as “Our study has some noteworthy limitations. First, small article scale about TLG and MTV didn’t support a high-quality meta-analysis, and our conclusion urgently need be further verified based on the large-scale prospective studies.”

Round 2

Reviewer 3 Report

In my previous review, I have addressed: "the main problem of the analysis is that the number of articles about MTV and TLG is few. The publication bias cannot be well assessed using funnel plot in such low number. It is doubtful that why TLG and MTV were still reported to be significant factors after the meta-analysis. Only the number of articles about SUVmax is somewhat enough to draw a conclusion." The authors did not revise the whole manuscript (including the abstract) according to my main suggestion although they have recognized the problem of publication bias and heterogeneity. They only responded to “other points need a further explanation”.  Because the number of publications about TLG, MTV, nodal SUVmax analyzed in this manuscript is low, it not suitable to write “Higher primary tumor total lesion glycolysis (TLG) (HR = 1.843, 95% CI = 1.100–-3.086, P = 0.02) and metabolic tumor volume (MTV) (HR = 2.06, 95% CI, 1.21–-3.51, P = 0.007) were significantly associated with shorter EFS.” in Abstract directly. The same flaw is for the description of nodal SUVmax.

The first paragraph of Discussion is also not well organized. The value of this manuscript was not well presented in this paragraph.

A concise summary at the end of the manuscript is lacking.

Overall speaking, this meta-analysis has some value (SUVmax for predicting OS). But the authors failed to address the strength and weaknesses of this meta-analysis in a proper way.

Author Response

Response to Reviewer 2 Comments

Point 1: In my previous review, I have addressed: "the main problem of the analysis is that the number of articles about MTV and TLG is few. The publication bias cannot be well assessed using funnel plot in such low number. It is doubtful that why TLG and MTV were still reported to be significant factors after the meta-analysis. Only the number of articles about SUVmax is somewhat enough to draw a conclusion." The authors did not revise the whole manuscript (including the abstract) according to my main suggestion although they have recognized the problem of publication bias and heterogeneity. They only responded to “other points need a further explanation”.  Because the number of publications about TLG, MTV, nodal SUVmax analyzed in this manuscript is low, it not suitable to write “Higher primary tumor total lesion glycolysis (TLG) (HR = 1.843, 95% CI = 1.100–-3.086, P = 0.02) and metabolic tumor volume (MTV) (HR = 2.06, 95% CI, 1.21–-3.51, P = 0.007) were significantly associated with shorter EFS.” in Abstract directly. The same flaw is for the description of nodal SUVmax.

Response 1: We are very sorry for this mistake, and we have revised whole paper and changed the Abstract as " Results: In total, 14 eligible studies with 1313 patients were included in this meta-analysis. Patients with high maximum standardized uptake value (SUVmax) have a shorter OS than those with low SUVmax (HR = 2.582, 95% CI 1.936-3.443, P < 0.001). Primary tumor SUVmax (HR = 1.938, 95% CI, 1.203-3.054, P = 0.004) were significantly correlated with EFS with relatively high heterogeneity (I2 = 84% and I2 = 69.4%, respectively). Based on the limited data, the combined HR for EFS with higher primary tumor total lesion glycolysis (TLG) and metabolic tumor volume (MTV) was 1.843(95% CI = 1.100-3.086, P = 0.02), and 2.06 (95% CI, 1.21-3.51, P = 0.007) respectively. Besides, the combined HR for OS with higher Nodal SUVmax was 2.095 (95% CI = 2.027-2.166, P < 0.001). Conclusion: High primary SUVmax has a significant association with the OS and EFS in patients treated with CCRT for LACC, which might serve as a prognostic predictor. For the limited data, the correlation about survival and TLG, MTV, nodal SUVmax need further large-scale prospective studies to explore. " Besides, we have revised the corresponding imprecise content in the full text.

Point 2: The first paragraph of Discussion is also not well organized. The value of this manuscript was not well presented in this paragraph.

Response 2: We appreciate your valuable comment and have reorganize the first paragraph of Discussion as" In this system review and meta-analysis, we firstly review the prognostic value of primary tumor SUVmax, SUVmean, MTV, TLG, and nodal SUVmax in patients treated with CCRT for LACC. Besides, the results of meta-analysis show patients with high primary SUVmax have a shorter OS and a higher risk of recurrence or progression than those with low SUVmax. The synthesized results showed that higher TLG and MTV were correlated with EFS and higher Nodal SUVmax was association with OS in patients treated with CCRT for LACC. For the limited data and existed publication bias and heterogeneity, whether MTV, TLG, and nodal SUVmax were significant marker and could act as a prognostic indicator in clinical need further large-scale prospective studies to explore. "

Point 3: A concise summary at the end of the manuscript is lacking.

Response 3: We are very sorry for this mistake and we have added this part as "

  1. Conclusion

High primary SUVmax has a significant association with the shorter OS and EFS in patients treated with CCRT for LACC, which might serve as a prognostic predictor. For the limited data and existed publication bias and heterogeneity, the correlation about survival and TLG, MTV, nodal SUVmax need further large-scale studies to explore. "

Point 4: Overall speaking, this meta-analysis has some value (SUVmax for predicting OS). But the authors failed to address the strength and weaknesses of this meta-analysis in a proper way.

Response 4: We are very sorry for the improper statement, and we have revised this part as: " Our study is the first system review and meta-analysis to evaluate the prognostic value of FDG PET/CT metabolic parameters (including primary-tumor SUVmax, SUVmean, MTV, TLG, and nodal SUVmax) in patients with LACC treated with CCRT. We performed an extensive search addressing all available databases, and our meta-analysis confirmed the significant association with primary-tumor SUVmax and OS of patients with LACC treated with CCRT. Future researches could focus on the prognosis-predicting model based on the primary-tumor SUVmax with using various types of prognostic factors. However, there still were some noteworthy limitations in this meta-analysis. First, small article scale about TLG and, MTV and SUVmax didn’t support a high-quality meta-analysis and valuable conclusion, and our conclusion urgently need be further verified based on the large-scale prospective studies were urgently need to explore the clinical value. Besides, all studies enrolled in the meta-analysis were retrospective, which might add selective bias. Third, significant heterogeneity existed in the results of this meta-analysis, and no subgroup analysis was performed for the limited studies. The potential source of the heterogeneity was probably from four aspects: (1) Patients in each study had different FIGO stages, different histologic subtypes, different treatment strategies, and different follow-up endpoints (Table 1-3); (2) the different PET/CT technique and image analysis were existed in each study, which might cause the heterogeneity in this analysis (Table S4); (3) the cut-off values for FDG PET/CT metabolic parameters were different in each study ; (4) the HR used in this meta-analysis was directly extracted from a Cox proportional hazards model (multivariate analysis and univariate analysis) or indirectly estimated from Kaplan-Meier curves."